# Positive and Negative Contribution from Lead–Oxygen Groups and Halogen Atoms to Birefringence: A First Principles Investigation

**DOI:** 10.3390/nano13233037

**Published:** 2023-11-28

**Authors:** Can Deng, Jialong Wang, Mei Hu, Xiuhua Cui, Haiming Duan, Peng Li, Ming-Hsien Lee

**Affiliations:** 1Xinjiang Key Laboratory of Solid State Physics and Devices, School of Physical Science and Technology, Xinjiang University, 777 Huarui Street, Urumqi 830017, China; dc202407@163.com (C.D.); 107552200765@stu.xju.edu.cn (J.W.); humei09292022@163.com (M.H.); dhm@xju.edu.cn (H.D.); 2Department of Physics, Tamkang University, New Taipei City 25137, Taiwan; mhslee@mail.tku.edu.tw

**Keywords:** first principle, oxyhalides, halogen, birefringence

## Abstract

Oxyhalides, containing oxygen and halogen atoms and combining the advantages of oxides and halides in geometry and optical response, have great potential in optical materials. In this study, the electronic structures and the optical properties of the Pb_3_O_2_X_2_ (X = Cl, Br, I) compounds have been investigated using the first principles method. The results show that these compounds have birefringence at 0.076, 0.078, and 0.059 @ 1064 nm, respectively. And, the asymmetric stereochemical active lone pair electrons were found around lead atoms, which were confirmed by the projected density of states, the electronic localization functions, and the crystal orbitals. The contribution of atoms and polyhedra to birefringence was further evaluated using the Born effective charge. The results show that halogen atoms give negative contribution, and lead—oxygen polyhedra give positive contribution. The spin—orbit coupling effect is also investigated, and the downshift of the conduction band and variation in the valence band are found after relevant spin—orbit coupling (SOC), which leads to a reduction in the band gap and birefringence.

## 1. Introduction

Birefringence is an optical phenomenon that was thought to be first discovered by Danish scientist Erasmus Bartolin in calcite crystals in 1669. When the light passes through certain materials like calcite, the light will split into two rays, and each ray has a different refractive index. Nowadays, birefringent materials are widely used in polarization devices, opto-isolators, circulators, beam displacers, and so on [1,2,3,4,5,6]. Except for a few minerals, more and more artificial birefringent materials have been discovered, including borates like K_3_B_6_O_10_Cl [7], phosphates like Sn_2_PO_4_X (X = Cl, Br, I) [8,9] and α-AZnPO_4_ (A = Li, K) [10], carbonates like LiMCO_3_ (M = K, Rb, Cs) [11] and CsPbCO_3_F [12], and metal oxyhalides like Sn_14_O_11_Br_6_ [13]. And certain kinds of birefringent compounds are still needed to meet the growing demands of optical communication, scientific research, and so on [14].

Oxyhalides, containing oxygen and halogen atoms and combining the advantages of oxides and halides in geometry and optical response, have great potential for the exploration of birefringent crystals, infrared window materials [15], and novel photocatalysts [16], examples including MoO_2_Cl_2_ [17,18], Ba_2_WO_3_F_4_ [19], AMoO_2_F_3_ (A = K, Rb, Cs, NH_4_, Tl) [20], and Pb_18_O_8_Cl_15_I_5_ [21]. It is interesting to note that introducing post-transition cations containing stereochemical active lone pairs was thought to be a good strategy for excellent birefringent compounds because stereochemical active lone pairs can lead to structural distortion and enhanced anisotropy polarization [22,23]. Examples of excellent birefringent compounds containing post-transition metal cations are Sn_2_B_5_O_9_X (X = Cl, Br) [24,25], Sn_9_O_4_Br_9_X (X = Cl, Br) [26], SbB_5_O_9_ [27], Pb_2_BO_3_Cl [28], etc. Hence, the authors believe the post-transition metal oxyhalides would have interesting optical performance.

The authors downloaded the Crystallographic Information Files (CIFs) of oxyhalides from ICSD and calculated their electronic structures and optical properties. And then, the very interesting positive and negative contributions coming from lead–oxygen polyhedra and halogen atoms to birefringence are found in Pb_3_O_2_X_2_ (X = Cl, Br, I) [29,30,31] compounds. In this paper, the authors report the electronic structure and the optical properties of Pb_3_O_2_X_2_ (X = Cl, Br, I) compounds. The results show that these compounds have moderate birefringence, and the asymmetric lone pair electrons are found around the lead atoms using the projected density of states, the electronic localization functions, and the crystal orbitals. The Born effective charges [32] show that the lead–oxygen polyhedra and the halogen atoms give positive and negative contributions to the total birefringence. The spin—orbit coupling effect is also investigated, and the downshift of the conduction band and the variation in the valence band are found after relevant spin—orbit coupling (SOC), which leads to a reduction in the band gap and birefringence.

## 2. Computational Details

In this paper, the electronic structures and the optical properties of the ternary lead oxyhalides Pb_3_O_2_X (X = Cl, Br, I) were investigated using the VASP code [33,34] based on density functional theory [35,36]. During the calculation, the Crystallographic Information Files (CIFs) of Pb_3_O_2_X (X = Cl, Br, I) compounds were obtained from the Inorganic Crystal Structure Database (ICSD, Version 5.1.0), which were Pb_3_O_2_Cl_2_ (ICSD-245918), Pb_3_O_2_Br_2_ (ICSD-245908), and Pb_3_O_2_I_2_ (ICSD-201857), respectively. The first principle calculations were performed using the Perdew—Burke—Ernzerhof (PBE) functional under the Generalized Gradient Approximation (GGA) [37,38]. The projector augmented wave (PAW) [39,40] formalism was adopted, and the valence electrons were set as Cl: 3*s*^2^3*p*^5^, Br: 4*s*^2^4*p*^5^, I: 5*s*^2^5*p*^5^, O: 2*s*^2^2*p*^4^, and Pb: 5*d*^10^6*s*^2^6*p*^2^. The kinetic energy cut-off was set as 400 eV, and the k-points of the Monkhorst-Pack grids of the Brillouin zone were set as Pb_3_O_2_Cl_2_ (2 × 4 × 3), Pb_3_O_2_Br_2_ (2 × 4 × 3), and Pb_3_O_2_I_2_ (1 × 4 × 3). Geometry optimization was performed, and the relaxation was stopped when the norms of all the forces were smaller than 0.02 eV/Å and the global break condition for the electronic SC loop was set as 1.0 × 10^−8^ eV. The obtained parameters of the Pb_3_O_2_X_2_ unit cell and the atomic coordinates are shown in Appendix A. By comparing the lattice vector changes before and after the DFT calculation, the relative error is less than 2%.

## 3. Results and Discussion

### 3.1. The Structures of Pb_3_O_2_X_2_ (X = Cl, Br, I)

Before investigating the electronic structures and the optical properties, the geometries of these compounds were first studied, because the geometry plays an important role in determining their properties. The geometries of these compounds are shown in Figure 1. As shown in Figure 1, they all crystallize in the orthorhombic space group *Pnma* with central symmetry. Although they have similar chemical formulas and the same space group, they have different functional basic units (FBUs). Take Pb_3_O_2_Cl_2_, for example, which consists of Pb(1)O_4_Cl_3_ polyhedra, Pb(2)O_2_Cl_2_ polyhedra, and Pb(3)O_2_Cl_4_ polyhedra; these FBUS are further interconnected into a three-dimensional structure by means of shared atoms. As for Pb_3_O_2_Br_2_, it consists of Pb(1)O_4_Br_3_, Pb(2)O_2_Br_4_, and Pb(3)O_2_Br_4_. And Pb_3_O_2_I_2_ consists of three different FBUs, which are Pb(1)O_4_I_3_, Pb(2)O_2_I_4_, and Pb(3)O_2_I, respectively.

It is well known that the post-transition metal atoms containing stereochemically active lone pairs would lead to structural distortion and an enhanced optical response. Hence, the authors investigate the distortion degree of these FBUs. The distortion degree is evaluated using Baur’s method implemented in VESTA [41] code. The Baur’s distortion index is defined as $D=\frac{1}{n}\sum_{i=1}^{n}\frac{\left|l_{i}-l_{av}\right|}{l_{av}}$ in which $l_{i}$ is the bond length and $l_{av}$ is the average bond length [42]. According to the definition of Baur’s distortion index, the distortion index of a regular polyhedron should be equal to zero. For example, the distortion indices of PO_4_ polyhedron in X_3_(PO_4_)_2_ were about 0.0017~0.0189 [32]. The obtained results of Pb_3_O_2_X (X = Cl, Br, I) compounds are shown in Table 1. As shown in Table 1, these FBUs have relatively large distortion indices, especially larger than the regular PO_4_ groups. The distorted FBUs are thought to be beneficial for enhanced birefringence.

### 3.2. The Electronic Structures of Pb_3_O_2_X_2_ (X = Cl, Br, I)

Using the method described above, the electronic structures and the refractive indices were calculated. Figure 2 shows the obtained GGA-PBE band structures at the high symmetry point of the irreducible Brillouin region. This result indicates that all three compounds mentioned above are indirect band gap compounds with band gap values of 2.30, 2.25, and 2.12 eV, respectively, noting that the GGA-PBE functional usually underestimates the band gap due to the discontinuity of exchange—correlation energy.

The obtained densities of states are also shown in Figure 2. As shown in Figure 2, all three of these compounds have similar projected density of states (PDOS). Take Pb_3_O_2_Cl_2_, for example. At the top of valence band, there are mainly O *p* states, Cl *p* states, and Pb *sp* states, and the main components of the bottom of the conduction band are Pb *sp* states along with O *p* and Cl *p* states. Furthermore, in the energy region in [−8, −5] eV below the Fermi level, one can find the hybrid states of Pb *s*–O *p*. And the hybrid states of Pb *p* and the antibonding states of (Pb *s*-O *p*)* are found in the energy region of [−1, 0] eV below the Fermi level. According to the revised model of lone pair electrons suggested by Walsh et al. [22], one can find the asymmetric lone pair electrons nearby the Fermi level (described below). After a detailed check of the PDOS, the authors believe the stereochemical activity of these lone pairs is very small, because they have very little Pb *p* states found near the Fermi level.

### 3.3. The Diagram of Asymmetric Lone Pairs in the Pb_3_O_2_X_2_ (X = Cl, Br, I)

The asymmetric lone pair electrons are further confirmed via the obtained electronic localized function (ELF) and the crystal orbitals. As shown in the left panel of Figure 3, the lobe-like or crescent ELF figure is found around the lead atoms (light yellow area), implying asymmetrically localized electrons can be found around the lead atoms, which should be the stereochemically active lone pairs of lead atoms. The asymmetric lone pair electrons are further confirmed via the crystal orbitals. The right panel gives the vision of crystal orbitals at the valence band maximum (VBM). One can clearly find the asymmetric orbitals localized around the lead atoms.

### 3.4. The Atomic Contribution to Optical Properties

Using the method described above, the refractive indices and the birefringence were obtained (shown in Figure 4). The results show that all these three compounds are biaxial crystals, and the obtained birefringence of Pb_3_O_2_Cl_2_, Pb_3_O_2_Br_2_, and Pb_3_O_2_I_2_ are 0.076, 0.078, and 0.059 @ 1064 nm, respectively. The obtained birefringence of Pb_3_O_2_Cl_2_ is similar to the experimental value (about 0.07) [43,44], and the obtained birefringence of Pb_3_O_2_X_2_ is slightly larger than the ab initio value shown in Ref. [43]. It is interesting to note that the birefringence of Pb_3_O_2_X_2_ is comparable with kinds of metal oxyhalides like KWO_3_F (0.088 @ 1064 nm) [45] and Pb_18_O_8_Cl_15_I_5_ (0.086 @ 636 nm) [21]. The moderate birefringence of Pb_3_O_2_X_2_ compounds may be related to the distorted polyhedra of FBUs and the asymmetric electron distribution of stereochemical lone pairs around lead atoms.

The atomic contributions were further investigated using the Born effective charges. The Born effective charge is defined as follows:$$q_{ij}^{Born}=\frac{\Omega }{e}\frac{\delta P_{i}}{\delta d_{j}}$$
in which the *δP_i_* is the change in polarization along the displacement direction *δd_j_*. According to the definition of the Born effective charges, the relatively large $q_{ij}^{Born}$ is beneficial to the relatively large polarization ${\delta P}_{i}$, hence the difference in the Born effective charges can reflect the difference in the polarization and then birefringence.

In this paper, the static dielectric matrix and the Born effective charges are calculated using the density functional perturbation theory implemented in VASP code. The obtained static dielectric matrix and the Born effective charges are given in Appendix A. Take Pb_3_O_2_Cl_2_, for example. The obtained macroscopic static dielectric tensors show that the tensors own the sequence as yy > xx > zz. For lead atoms, the Born effective charges own the sequence as q_yy_ > q_xx_ > q_zz_ (for Pb1), q_yy_ > q_zz_ > q_xx_ (for Pb2), and q_yy_ > q_xx_ > q_zz_ (for Pb3). The birefringence is the difference of the largest and the smallest refractive indices, so we would use the difference in the Born effective charges $\Delta q=q_{yy}-q_{zz}$ to reflect the atomic contribution to birefringence. The obtained diagonal elements and the difference in the Born effective charges Δq are shown in Appendix A. If the difference in the Born effective charges has a similar sign to the nominal charge, the atoms give positive contribution to birefringence; otherwise, they give negative contribution to birefringence. The differences in the Born effective charges Δq of lead atoms are 1.08, 0.37, and 0.29, so the lead atoms give a positive contribution to birefringence. The difference in the Born effective charges Δq of the Cl2 atom is 0.28, indicating that it gives a negative contribution to the total birefringence. Similar conclusions can also be drawn in two other compounds, noting that, except the diagonal elements of the Born effective charges, the nondiagonal elements of the Born effective charges also give a contribution to the total birefringence.

### 3.5. The Band Structures and Birefringence Induced by SOC

It is well known that the band structures and the optical properties could be affected by the SOC coming from heavy atoms like lead and bismuth atoms. For example, E. Narsimha Rao et al. pointed out that, for the CsPbCO_3_F compound [46], the obtained band gaps with SOC are 3.03 (PBE), 2.94 (PBEsol), and 4.45 (TB-mBJ), while the band gaps without SOC are 3.59 (PBE), 3.48 (PBEsol), and 5.58 (TB-mBJ), respectively. In this paper, the authors have also investigated the SOC effect on the band structures and the birefringence of Pb_3_O_2_Cl_2_ compounds. The obtained band structures, with and without relevant SOC, are given in Appendix A. As shown in Appendix A, the obtained band gap with relevant SOC is about 2.16 eV, smaller than the band gap of 2.30 eV without relevant SOC. The reduction in the band gap comes from the downshift of the conduction bands and the changing of the top of the valence band. The changing of the band structures induced by SOC leads to variation in the refractive indices and birefringence. The obtained refractive indices and birefringence with and without relevant SOC are given in Figure 5. The obtained refractive indices with relevant SOC are about 0.069 @ 1064 nm, smaller than the value without relevant SOC (about 0.076 @ 1064 nm). It is worth noting that the birefringence with relevant SOC is similar to the experimental value.

## 4. Conclusions

In this paper, the electronic structures and the optical properties of a series of lead oxyhalides Pb_3_O_2_X_2_ (X = Cl, Br, I) were investigated using the first principles method. The results show that these compounds have relatively small birefringence, which are 0.076, 0.078, and 0.059 @ 1064 nm, respectively. These compounds have asymmetric stereochemically active lone pair electrons around lead atoms, which can be confirmed by the projected density of states, electronic localization functions, and crystal orbitals. The Born effective charges show that the lead–oxygen polyhedra give a positive contribution and halogen gives a negative contribution to the total birefringence. The spin—orbit coupling effect was also investigated, and the downshift of the conduction band and the variation in the valence band were found after relevant SOC, which led to a reduction in the band gaps and birefringence.

## Figures and Tables

**Figure 1 nanomaterials-13-03037-f001:**
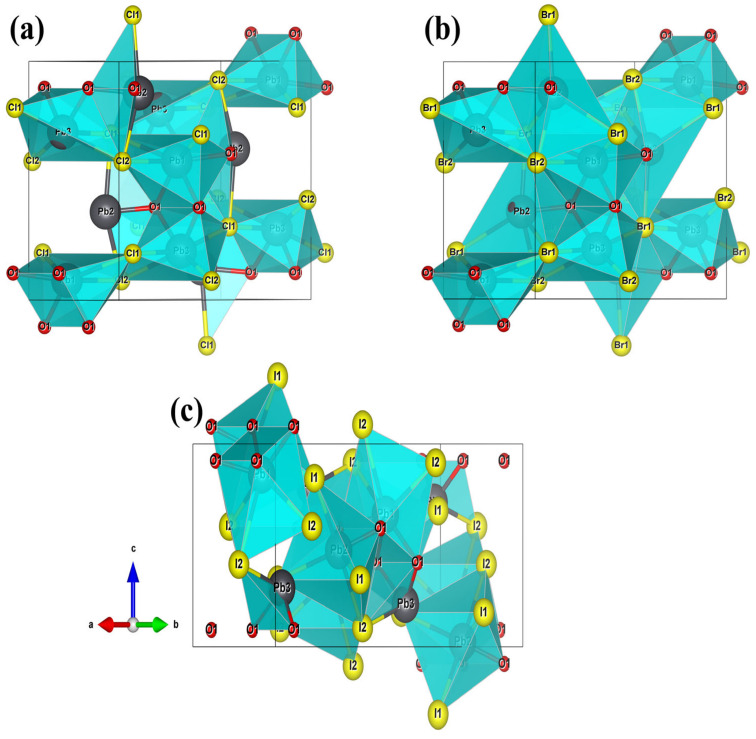
Crystal structures of (**a**) Pb_3_O_2_Cl_2_, (**b**) Pb_3_O_2_Br_2_, and (**c**) Pb_3_O_2_I_2_ (Pb: Black, O: Red, Halogens: Yellow).

**Figure 2 nanomaterials-13-03037-f002:**
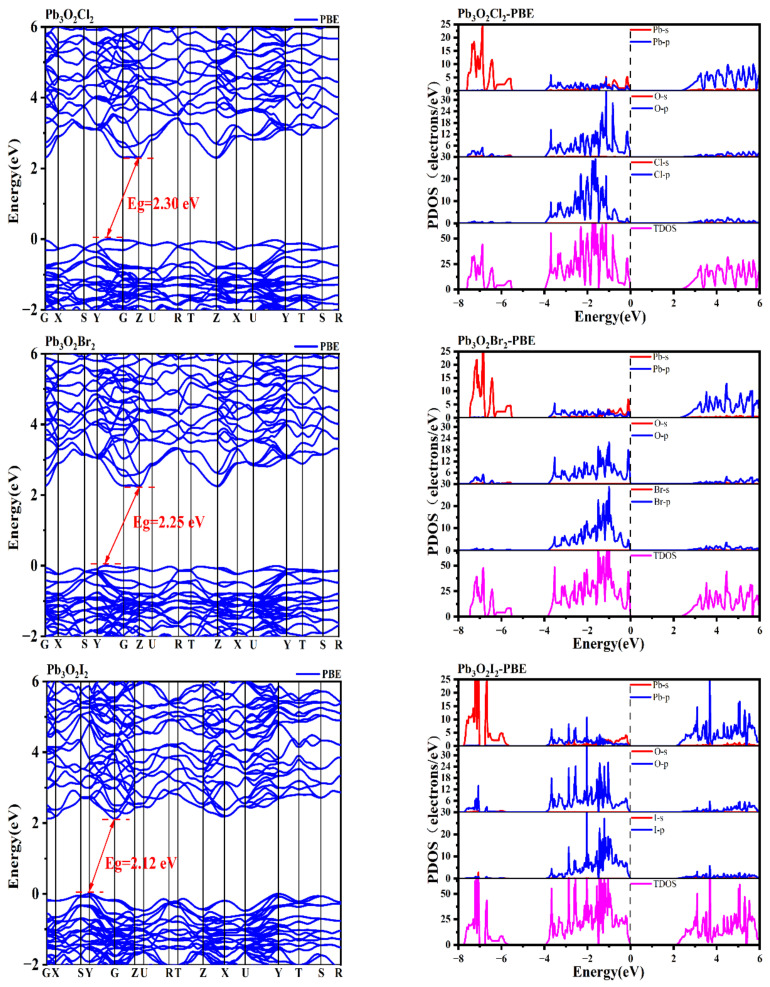
Band structures and the projected density of states (PDOS) of Pb_3_O_2_X_2_ (X = Cl, Br, I).

**Figure 3 nanomaterials-13-03037-f003:**
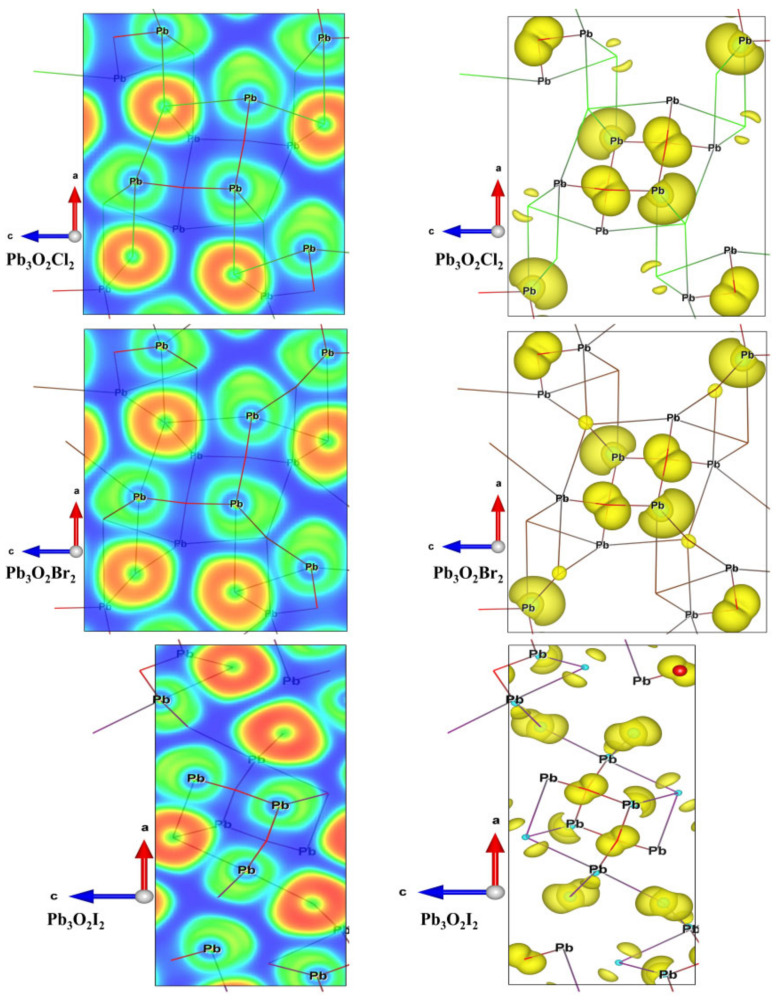
Electronic Localized Function (ELF, left, Blue for the electron off-domain state, other colors for the electron localized state) and the crystal orbitals (right, Yellow represents the distribution of electrons in orbitals near the Fermi surface) of Pb_3_O_2_Cl_2_, Pb_3_O_2_Br_2_, and Pb_3_O_2_I_2_.

**Figure 4 nanomaterials-13-03037-f004:**
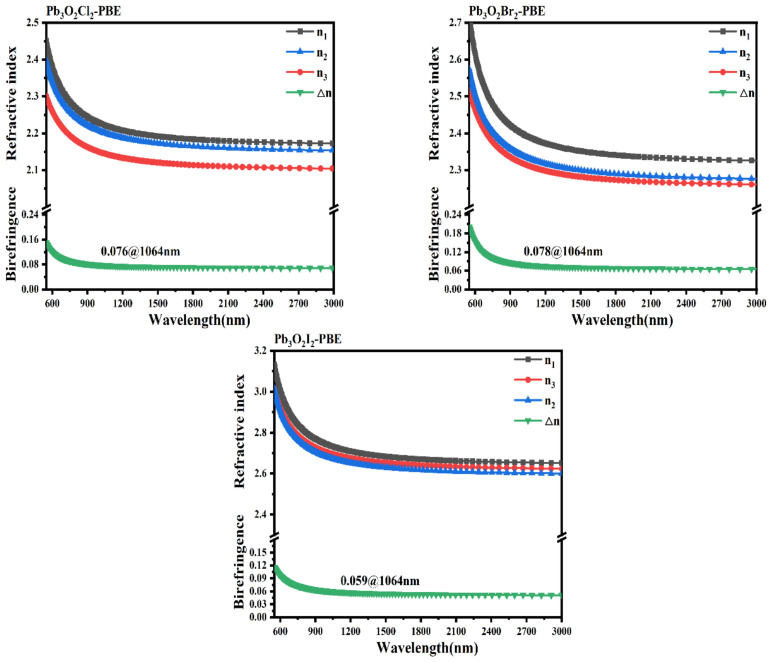
The refractive indices and birefringence of Pb_3_O_2_X_2_ (X = Cl, Br, I).

**Figure 5 nanomaterials-13-03037-f005:**
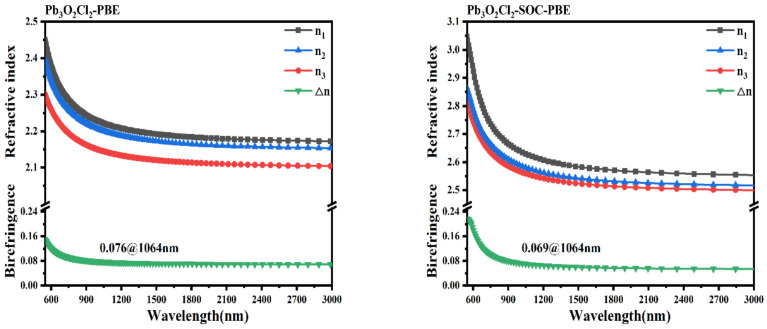
The refractive indices and birefringence with (**left**) and without (**right**) relevant SOC of Pb_3_O_2_Cl_2_.

**Table 1 nanomaterials-13-03037-t001:** The distortion indices of Pb_3_O_2_X_2_ (X = Cl, Br, I) compounds obtained using Baur’s method.

**Pb_3_O_2_Cl_2_**	Group	Pb(1)O_4_Cl_3_	Pb(2)O_2_Cl_2_	Pb(3)O_2_Cl_4_
	Distortion index	0.16923	0.15066	0.13839
**Pb_3_O_2_Br_2_**	Group	Pb(1)O_4_Br_3_	Pb(2)O_2_Br_4_	Pb(3)O_2_Br_4_
	Distortion index	0.18539	0.16644	0.14621
**Pb_3_O_2_I_2_**	Group	Pb(1)O_4_I_3_	Pb(2)O_2_I_4_	Pb(3)O_2_I
	Distortion index	0.19859	0.16010	0.17105

## Data Availability

The data that support the findings of this study are available from the corresponding author upon reasonable request.

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
