# Peer review of "Positive and Negative Contribution from Lead–Oxygen Groups and Halogen Atoms to Birefringence: A First Principles Investigation"

_nanomaterials, 2023, doi:10.3390/nano13233037_

Round 1
Reviewer 1 Report
Comments and Suggestions for Authors
The manuscript presents results on the electronic structures and optical properties of the Pb3O2X2 (X = Cl, Br, I) compounds investigated using the first-principles method. The results are new and interesting. My biggest complaint concerns the lack of comparison of the obtained computational data with the corresponding results for other lead-containing materials (experimental and theoretical). Such a comparison should be included in the discussion or conclusions of the work. The reader should be able to analyze the comparison of appropriate parameters or indices for various lead-containing materials, especially lead-oxygen optical materials.
I also have a few technical comments: Figures 1 and 3 should be enlarged because they are illegible; all tables are too large, they have too large paragraphs. What is accuracy of distortion indices and effective charge values (Table 1 and 3)? Please check the meaning of entering 5 digits after the decimal point.
After correction, the manuscript will be ready for publication
Comments on the Quality of English LanguagePlease improve the language, style and grammar (lines 26, 66, 80, 101, 108). Colloquialism is inappropriate in scientific work, e.g. "and so on..., successful stories..., oxyhalides like Sn14O11Br6 [13], et al., let's check"
After correction, the manuscript will be ready for publication
Reviewer 2 Report
Comments and Suggestions for Authors
In this work, three lead oxyhalides are studied on the basis of first-principles calculations.
The electronic structure, which has been calculated before, has some differences with previous calculations which are not discussed.
Then the authors present the electron localization function and crystal orbitals, which shows the asymmetric density due to the Pb electron lone pair, as expected.
Concerning the birefringence, the main object of the study, the reported values are small.
Unfortunately I cannot recommend the paper for publication, mainly because I think the results do not have enough significance and novelty to be published in Nanomaterials.
The manuscript also has several flaws, and the discussion could be improved. I believe fhe following comments would improve the manuscript if taken into account.
44 "The authors have downloaded the crystallographic information files of oxyhalides from ICSD and Materials Project database"
In the methods only ICSD structures are given, which one is it?
If materials project is used it should be added in the methods section, if not it should be removed here.
But I suggest the authors remove this information from the introduction and leave it only in the methods since it's more a technical thing.
The discussion on the structure is incomplete. Baur's indices should be explained to understand how it is relatively large, and relative to what.
The Materials Project already contains electronic structure data for these compounds. In this database, the band gap for the X=Cl compound is 2.43 eV, compared to 2.73 eV in this work. Furthermore, in that database the band structure shows an indirect band gap, as opposed this work which shows a direct band gap. Similar differences can be seen for the X=Br compound. In the X=I compound the MP band gap is also smaller, although both calculations result in an indirect band gap. Can these differences be ascribed to the different structure used (ICSD)? What are the stresses and atomic forces? This should be checked, as a check of the quality of the calculation, to be sure the theoretical structure be different from the experimental? I also think the crystal structures used should be displayed in the article, to aid reproducibility of the calculations by others.
This discussion would improve the the new calculations by distinguishing them from the existing (MP) calculations of the same properties, and the difference in properties would be clarified.
108 "The results indicates that the FBUs of Pb-O-Cl give main contribution in determining the bandgap and refractive indices."
Isn't every atom in the cell part of FBUs of Pb-O-Cl? It seems that this sentence means nothing.
What does the "origin" row mean in table 2? I think Origin should be above the rows, naming the column with the different groups, and the rows named "origin" should be "Total" instead.
166 "Δq is defined as the difference between the maximum value, qzz and the minimum, qyy."
Either this definition is incorrect, or the table is. Looking at the values, it should be qxx-qzz.
Can the authors give an explanation of the RSAC method used, and expand on the relation between BECs and birefringence, or give a useful reference (see below)?
Typos:
45 database - databases
48 Compounds - compounds
64 calculation - calculations
70 convergent - convergence
55 birefringence comparison - birefringence in comparison
110 found - find
18 "electronic localized functions" should be changed to electronic localization functions wherever it appears in the text.
Table 3: "Elemental" should be atomic site, or site, or similar.
52 The real-spacing atom-cutting (RSAC) [32] results and Born effective charges [33]
Ref. 33 does not discuss Born effective charges, so it is not correct.
83 PbO2Br4, and PbO2Br4
Repeated FBUs, one of these must be different.
References
33 Not Phys. Rev. B but Physica B
Ref. 39 is not about GGA, so it should not be in line 65 as "Generalized Gradient Approximation (GGA) [39]"
The english is understandable, in general. Some typos and changes in technical terms are given previously.
Reviewer 3 Report
Comments and Suggestions for Authors
Authors studied series of lead oxyhalides Pb3O2X2 (X = Cl, Br, I) using periodic DFT method and calculated electronic and optical properties.
Data are displayed in 4 Tables and in 3 Figures.
Minor issues:
l.26-32: please extend the introductory text with some more concrete examples. When was birefrigences first discovered, what was the very first material in an optical device ?
l.65 "The core and valence electrons are treated using Norm-Conserving Pseudopotentials (NCP)" ... please be more precise - core electrons are treated using pesudopotentials. But valence electrons using plane-waves, according to the CASTEP program description.
But here is the serious issue: relativistic effects are active beacuse of the Pb atom. Please specify whether used NCP pseudopotential covers scalar relativistic effects (I assume yes). But how about the spin-orbital effects ? What is their influence on calculated properties - the refractive indices and birefringence ? See for example articles, https://ui.adsabs.harvard.edu/abs/2017SuMi..110..221Z/abstract , https://doi.org/10.1063/5.0146397 , https://pubs.rsc.org/en/content/articlelanding/2017/cp/c7cp05750d/unauth .
Therefore calculate the properties with spin-orbital effects and extend the manuscript accordingly.
Reviewer 4 Report
Comments and Suggestions for Authors
This work presents a computational study on the electronic structures and optical properties the of Pb3O2X2 (X = Cl, Br, I) based on first-principles calculations. In this work, a relatively small birefringence for these systems were obtained, and the asymmetric stereochemical active lone pair electrons around lead atoms were attributed to the source of their such small birefringence. Moreover, the atomic and polyhedra’ contributions to birefringence were further examined. Overall, this work lacks novelty due to the similarity of previously published work for Pb3O2X2 (X = Cl, Br, I) (see Zakiryanov et al. Physics of the Solid State, 2017, 59, 4, 710–721). Furthermore, the birefringence results obtained for these systems are inconsistent with literature. Additionally, the state-of-the-art of modeling Pb-based systems was not followed. Thus, I thus do not recommend its publication. Below are my comments.
Comment#1: The reported birefringence values are 0.05 (exp: 0.07), 0.03, 0.04 for Pb3O2X2 (X = Cl, Br, I) (see Zakiryanov et al. Physics of the Solid State, 2017, 59, 4, 710–721). The values reported in this work are in the opposite order and inconsistent with the measured value for Pb3O2Cl2.
Comment#2: The DFT-optimized and measured lattices parameters need to be compared.
Comment#3: Pb is a heavy atom. The spin-orbit coupling (SOC) effect is present in Pb-X systems. The contributions from the lone electron pair around the Pb atom could be different due to the SOC effect. So, the SOC effect also needs to be incorporated in the DFT calculations to properly demonstrate the effect of the lone electron pair around Pb.
Comment#4: Figure 3: It lacks the professional quality and needs to be reproduced. Figures a,b,c are almost overlapped, hard to distinguish one from the other. Also, keep the two panels of Figures, a,b,c and d,e,f at a reasonable distance and consider increasing the font size of the color bars.

This quality of English in this manuscript is okay.
Round 2
Reviewer 1 Report
Comments and Suggestions for Authors
After making all necessary corrections to the manuscript, the paper can be published.
Reviewer 2 Report
Comments and Suggestions for Authors
The authors have correctly replied to my comments. Then new calculations shows better values of birefringence and electronic properties. The manuscript has changed sufficiently for the better, and I now support its publication.
Comments on the Quality of English LanguageEnglish language is fine.
Reviewer 3 Report
Comments and Suggestions for Authors
All my comments were accepted, now publish as such.
Reviewer 4 Report
Comments and Suggestions for Authors
In the revised manuscript, the authors have addressed my concerns and improved significantly. They have corrected their methodology and obtained results consistent with the experiment. Therefore, I recommend its publications in the present form.
Comments on the Quality of English LanguageOverall, the quality of English is at a satisfactory level. The choice of words and sentence-making can be further improved for better readability and appropriateness.
For example, in the last sentence of the Abstract, the phrase '.......after SOC concerning....' does not sound appropriate. Consider writing something like '........after incorporating the SOC effect,.......'. or 'due to the due to SOC effect,....'
Also, in the Abstract, the acronym SOC is used, but it was not defined earlier. Define if first.